# Learning Symmetric Rules with SATNet

**Sangho Lim**[*]
School of Computing
KAIST
Daejeon, South Korea
lim.sang@kaist.ac.kr

**Eun-Gyeol Oh**[*]
Graduate School of Information Security
KAIST
Daejeon, South Korea
eun-gyeol.oh@kaist.ac.kr

**Hongseok Yang**
School of Computing and Kim Jaechul Graduate School of AI, KAIST
Discrete Mathematics Group, Institute for Basic Science (IBS)
Daejeon, South Korea
hongseok.yang@kaist.ac.kr

## Abstract

SATNet is a differentiable constraint solver with a custom backpropagation algorithm, which can be used as a layer in a deep-learning system. It is a promising proposal for bridging deep learning and logical reasoning. In fact, SATNet has been successfully applied to learn, among others, the rules of a complex logical puzzle, such as Sudoku, just from input and output pairs where inputs are given as images. In this paper, we show how to improve the learning of SATNet by exploiting symmetries in the target rules of a given but unknown logical puzzle or more generally a logical formula. We present SymSATNet, a variant of SAT-Net that translates the given symmetries of the target rules to a condition on the parameters of SATNet and requires that the parameters should have a particular parametric form that guarantees the condition. The requirement dramatically reduces the number of parameters to learn for the rules with enough symmetries, and makes the parameter learning of SymSATNet much easier than that of SATNet. We also describe a technique for automatically discovering symmetries of the target rules from examples. Our experiments with Sudoku and Rubik's cube show the substantial improvement of SymSATNet over the baseline SATNet.

## 1 Introduction

Bringing the ability of reasoning to the deep-learning systems has been the aim of a large amount of recent research efforts [28, 14, 5, 26, 23]. One notable outcome of these endeavours is SATNet [26], a differentiable constraint solver with an efficient custom backpropagation algorithm. SATNet can be used as a component of a deep-learning system and make the system capable of learning and reasoning about sophisticated logical rules. Its potential has been demonstrated successfully with the tasks of learning the rules of complex logical puzzles, such as Sudoku, just from input-output examples where the inputs are given as images.

We show how to improve the rule (or constraint) learning of SATNet, when the target rules have permutation symmetries. By having symmetries, we mean the solutions of the rules are closed under the permutations of those symmetries. For example, in Sudoku, if a completed $9 \times 9$ Sudoku board is a solution, permuting the numbers 1 to 9 in the board, the first three rows, or the last three columns always gives rise to another solution. Thus, these permutations are symmetries of Sudoku.

---

[*]These authors contributed equally to this work.

36th Conference on Neural Information Processing Systems (NeurIPS 2022).

Our improvement is a variant of SATNet, called SymSATNet, which abbreviates symmetry-aware SATNet. SymSATNet assumes that some symmetries of the target rules are given a priori although the rules themselves are unknown. It then translates these symmetries into a condition on the parameter matrix $C \in \mathbb{R}^{n \times n}$ of SATNet (or our minor generalisation), and requires that the parameters have a particular parametric form that guarantees the condition. Concretely, the translated condition says that the matrix $C$ regarded as a linear map should be equivariant with respect to the group $G$ determined by the given symmetries, and the requirement is that $C$ should be a linear combination of elements in a basis for the space of $G$-equivariant symmetric matrices. The coefficients of this linear combination are the parameters of our SymSATNet, and their number is often substantially smaller than that of the parameters of SATNet.[1] For Sudoku, the former is 18, while the latter is $729^2$ or $k \cdot 729$ for some $k \in \mathbb{N}$ at best. The reduced number of parameters implies that SymSATNet has to tackle an easier learning problem than SATNet, and has a potential to learn faster and generalise better than SATNet.

Who provides symmetries for SymSATNet? The default answer is domain experts, but a better alternative is possible. We present an automatic algorithm to discover symmetries. Our algorithm is based on empirical observation that symmetries emerge in the parameter matrix $C$ of SATNet in the early phase of training, as clusters of similar entries. Our algorithm takes a snapshot of $C$ at some training epoch of SATNet, and finds a group $G$ such that (i) specific entries of $C$ share similar values by $G$-equivariance condition, and (ii) the number of SymSATNet parameters under $G$ is minimised.

We empirically evaluate SymSATNet and our symmetry-discovering algorithm with Sudoku and a problem related to Rubik's cube. For both problems, our algorithm discovered nontrivial symmetries, and SymSATNet with manually specified or automatically found symmetries outperformed the baseline SATNet in learning the rules, in terms of both efficiency and generalisation.

**Related work** There have been multiple studies on discovering symmetries present in conjunctive normal form (CNF) formulas in order to reduce the search space of satisfiability (SAT) solvers. Crawford [7] proved that the symmetry-detection problem is equivalent to the graph isomorphism problem, and showed how to reduce the complexity of pigeonhole problems using symmetries. Crawford et al. [8] proposed symmetry-breaking predicates (SBPs), and Aloul et al. [1, 2] developed SBPs with more efficient constructions. For automatic symmetry detection, Darga et al. [9] presented a method that improves the partition refinement procedure introduced by McKay [19, 20], and Darga et al. [10] proposed an algorithm that achieve efficiency by exploiting the sparsity of symmetries. In contrast to these global and static methods, Benhamou et al. [4] and Devriendt et al. [12] handled local symmetries that dynamically arise during search. The use of symmetries also appears in NeuroSAT [22], which learns how to solve SAT problems from examples. NeuroSAT solves a given SAT formula by message passing over a graph constructed from the formula, and in so doing, its learnable solver can exploit symmetries in the formulas. All of these techniques use symmetries to help solve given formulas, whereas our approach uses symmetries to help learn such formulas. Another difference is that those techniques find hard symmetries of given formulas, whereas our approach discovers soft or approximate symmetries in a given SATNet parameter matrix. In the context of deep learning, Basu et al. [3] and Dehmamy et al. [11] described algorithms that find and exploit symmetries via group decompositions and Lie algebra convolutions. But these techniques are not designed to find symmetries in logical formulas.

Our work is related to the studies on learning logical rules from examples using gradients. Yang et al. [28] proposed neural logic programming, an end-to-end differentiable system which learns first-order logical rules, Evans and Grefenstette [14] proposed a differentiable inductive logic programming system which is robust to noise of training data, and Cingillioglu and Russo [5] introduced an RNN-based model to learn logical reasoning tasks end-to-end. Want et al. [25, 26] presented SATNet using the mixing method, and Topan et al. [23] further improved SATNet by solving the symbol grounding problem, a key challenge of SATNet. Our work extends these lines of work by proposing how to discover and exploit symmetries from examples when learning logical rules with SATNet.

---

[1]SATNet assumes that $C$ is of the form $S^T S$ for some $S \in \mathbb{R}^{m \times n}$ for $m < n$, so that the number of parameters is $mn < n^2$. But often it is still substantially larger than the number of parameters of SymSATNet.

## 2 Background

We review SATNet, the formalisation of symmetries using groups, and equivariant maps. For a natural number $n$, let $[n] = \{1, \ldots, n\}$, and for a matrix $M$, let $M_{i,j}$ be the $(i,j)$-th entry of $M$.

### 2.1 SATNet

A good starting point for learning about SATNet is to look at its origin, the mixing method [25], which is an efficient algorithm for solving semidefinite programming problems with diagonal constraints. Let $n, k \in \mathbb{N}$ and $C$ be a real-valued symmetric matrix in $\mathbb{R}^{n \times n}$. The mixing method aims at solving the following optimisation problem:

$$\operatorname*{argmin}_{V \in \mathbb{R}^{k \times n}} \langle C, V^T V \rangle \quad \text{subject to } \|v_i\| = 1 \text{ for } i \in [n] \tag{1}$$

where $v_i$ is the $i$-th column of the matrix $V$, and $\|v_i\|$ is $L_2$ norm of $v_i$. The mixing method solves (1) by coordinate descent, where each column $v_i$ of $V$ is repeatedly updated as follows: $g_i \leftarrow \sum_{j \in [n], j \neq i} C_{i,j} v_j$ and $v_i \leftarrow -\frac{g_i}{\|g_i\|}$. This always finds a fixed point of the equations. In fact, it is shown that almost surely this fixed point attains a global optimum of the optimisation problem.

An example of the above optimisation problem most relevant to us is a continuous relaxation of MAXSAT. MAXSAT is a problem of finding truth assignments to $n$ boolean variables $b_1, \ldots, b_n$. It assumes $m$ clauses of those variables, $F_1, \ldots, F_m$, where $F_\ell$ is the disjunction of some variables with or without negation: $F_\ell = b_{i_1} \vee \ldots \vee b_{i_p} \vee \neg b_{i_{p+1}} \vee \ldots \neg b_{i_{p+q}}$. Then, MAXSAT asks for a truth assignment on the variables that maximises the number of true clauses $F_i$ under the assignment. The rules of many problems, including Sudoku, can be expressed as an instance of MAXSAT.

To apply the mixing method to MAXSAT, we introduce relaxed vectors $v_1, \ldots, v_n \in \mathbb{R}^k$ that encode the boolean variables, and construct the matrix $S \in \mathbb{R}^{m \times n}$ that encodes the $m$ clauses of MAXSAT: the $(\ell, j)$-th entry of $S$ has 1 if $F_\ell$ contains $b_j$; and $-1$ if $F_\ell$ includes $\neg b_j$; and 0 if neither of these cases holds. Then, the problem in (1) is formed with $C = -S^T S$, and solved by the mixing method.

SATNet is a variant of the mixing method where some of the columns of $V$ are fixed and the optimisation is over the rest of the columns.[2] Concretely, it assumes that the column indices in $[n]$ are split into two disjoint sets, $\mathcal{I}$ and $\mathcal{O}$ (i.e., $\mathcal{I} \cup \mathcal{O} = [n]$ and $\mathcal{I} \cap \mathcal{O} = \emptyset$). The inputs of SATNet are the columns $v_i$ of $V$ with $i \in \mathcal{I}$, and the outputs are the rest of the columns (i.e., the $v_o$'s with $o \in \mathcal{O}$). The symmetric matrix $C$ is the parameter of SATNet. Given the input vectors, SATNet repeatedly executes the coordinate descent updates on each output column, until it converges.

One important feature of SATNet is that it has a custom algorithm for backpropagation. Let $V_{\mathcal{I}}$ be the matrix of the input columns to SATNet, and $V_{\mathcal{O}}$ be that of the output columns computed by SATNet on the input $V_{\mathcal{I}}$ under the parameter $C$. Assume that $l$ is a loss of the output $V_{\mathcal{O}}$. In this context, SATNet provides formulas and algorithms for computing the derivatives $\partial l / \partial V_{\mathcal{I}}$ and $\partial l / \partial C$.

We recall the formulas for the derivatives. Let $o_1 < o_2 < \ldots < o_{|\mathcal{O}|}$ be the sorting of the indices in $\mathcal{O}$. Assume that SATNet was run until convergence, so that the output columns in $V_{\mathcal{O}}$ are the fixed point of the coordinate descent updates: for all $o \in \mathcal{O}$, $g_o = \sum_{j \in [n], j \neq o} C_{o,j} v_j$ and $v_o = -\frac{g_o}{\|g_o\|}$. The formulas for $\partial l / \partial V_{\mathcal{I}}$ and $\partial l / \partial C$ at $(V_{\mathcal{I}}, V_{\mathcal{O}}, C)$ are defined in terms of the next quantities:

$$C', D' \in \mathbb{R}^{|\mathcal{O}| \times |\mathcal{O}|}, \qquad P \in \mathbb{R}^{|\mathcal{O}|k \times |\mathcal{O}|k}, \qquad U \in \mathbb{R}^{|\mathcal{O}|k \times 1}, \qquad W \in \mathbb{R}^{|\mathcal{O}|k \times n^2}.$$

They have the following definitions: for $i, j \in [|\mathcal{O}|]$, $p, q \in [k]$, and $r, s \in [n]$,

$$(C')_{i,j} = \begin{cases} 0 & \text{if } i = j \\ C_{o_i, o_j} & \text{if } i \neq j, \end{cases} \qquad U = (P((D' + C') \otimes I_k))^\dagger \left( \frac{\partial l}{\partial \operatorname{vec}(V_{\mathcal{O}})} \right)^T,$$

$$(D')_{i,j} = \begin{cases} \|g_{o_i}\| & \text{if } i = j \\ 0 & \text{if } i \neq j, \end{cases}$$

$$P_{\substack{(i-1)k+p, \\ (j-1)k+q}} = \begin{cases} (I_k - v_{o_i} v_{o_i}^T)_{p,q} & \text{if } i = j \\ 0 & \text{if } i \neq j, \end{cases} \qquad W_{\substack{(i-1)k+p, \\ (r-1)n+s}} = \begin{cases} 0 & \text{if } r = o_i \text{ and } s = o_i \\ V_{p,s} & \text{if } r = o_i \text{ and } s \neq o_i \\ V_{p,s} & \text{if } r \neq o_i \text{ and } s = o_i \\ 0 & \text{if } r \neq o_i \text{ and } s \neq o_i. \end{cases}$$

---

[2] The original SATNet assumes that $C$ has the form $S^T S$ for some $m \times n$ matrix $S$. We drop this assumption and adjust the forward and backward computations of SATNet accordingly. The main steps of derivations of the formulas for the forward and backward computations are from the work on SATNet [26].

Here $\otimes$ is the Kronecker product, $\_^\dagger$ is Moore-Penrose inverse (also known as pseudo inverse), and $\mathrm{vec}(V_\mathcal{O})$ is the vector obtained by stacking the columns of $V_\mathcal{O}$. Let $C_{\mathcal{O},\mathcal{I}} \in \mathbb{R}^{|\mathcal{O}| \times |\mathcal{I}|}$ be obtained by restricting $C$ to the indices $(o, i)$ with $o \in \mathcal{O}$ and $i \in \mathcal{I}$. Then,

$$\partial l / \partial \, \mathrm{vec}(C) = -U^T W, \qquad \partial l / \partial \, \mathrm{vec}(V_\mathcal{I}) = -U^T(C_{\mathcal{O},\mathcal{I}} \otimes I_k). \tag{2}$$

SATNet computes the above derivative formulas efficiently by iterative algorithms.

## 2.2 Symmetries and equivariant maps

By symmetries on a set $\mathcal{X}$, we mean a group $G$ that acts on $\mathcal{X}$. The acting here refers to a function $\_ \cdot \_$ from $G \times \mathcal{X}$ to $\mathcal{X}$, called *group action*, such that (i) $e \cdot x = x$ for the unit $e \in G$ and any $x \in \mathcal{X}$, and (ii) $(g \circ g') \cdot x = g \cdot (g' \cdot x)$ for all $g, g' \in G$ and $x \in \mathcal{X}$, where $\_ \circ \_$ is the group operator of $G$.

We use symmetries of permutations on a finite set. The set $\mathcal{X}$ in our case is $\mathbb{R}^{k \times n}$, the space of the matrix $V$ in (1), and $G$ is a subgroup of the group $\mathcal{S}_n$ of all permutations on $[n]$. The group action $g \cdot V$ is then defined by permuting the columns of $V$ by $g$: for all $i, j \in [n]$, $(g \cdot V)_{i,j} = V_{i,g^{-1}(j)}$. This group action can be expressed compactly with the $n \times n$ permutation matrix $P_{g^{-1}}$ where $(P_{g^{-1}})_{i,j} = \mathbb{1}_{\{i = g^{-1}(j)\}}$ and $g \cdot V = V P_{g^{-1}}$ for the indicator function $\mathbb{1}$. Throughout the paper, we often equate each element $g$ of $G$ with its permutation matrix $P_g$, and view $G$ itself as the group of permutation matrices $P_g$ for $g \in G$ with the standard matrix multiplication.

One important reason for considering symmetries is to study maps that preserve these symmetries, called equivariant maps. Let $G$ be a group that acts on sets $\mathcal{X}$ and $\mathcal{Y}$.

**Definition 2.1.** A function $f : \mathcal{X} \to \mathcal{Y}$ is *G-equivariant* or *equivariant* if $f(g \cdot x) = g \cdot (f(x))$ for all $g \in G$ and $x \in \mathcal{X}$. It is *G-invariant* or *invariant* if $f(g \cdot x) = f(x)$ for all $g \in G$ and $x \in \mathcal{X}$.

The forms of equivariant maps have been studied extensively in the work on equivariant neural networks and group representation theory [6, 18, 27]. In particular, when $f$ is linear, various representation theorems for different $G$'s describe the matrix form of $f$. We use permutation groups defined inductively by the following three operations.

**Definition 2.2.** Let $G$ and $H$ be permutation groups on $[p]$ and $[q]$, with each group element viewed as a $p \times p$ or $q \times q$ permutation matrix. The *direct sum* $G \oplus H$, the *direct product* $G \otimes H$, and the *wreath product* $H \wr G$ are the following groups of $(p+q) \times (p+q)$ or $pq \times pq$ permutation matrices with matrix multiplication as their composition:

$$G \oplus H = \{g \oplus h : g \in G, h \in H\}; \qquad G \otimes H = \{g \otimes h : g \in G, h \in H\};$$

$$H \wr G = \{\mathrm{wr}(\vec{h}, g) : g \in G, \vec{h} \in H^p\}.$$

Here $g \oplus h$ is the block diagonal matrix with $p \times p$ matrix $g$ as its upper-left corner and $q \times q$ matrix $h$ as the lower-right corner, $g \otimes h$ is the Kronecker product of two matrices $g$ and $h$, and $\mathrm{wr}(\vec{h}, g)$ is the $pq \times pq$ permutation matrix defined by $\mathrm{wr}(\vec{h}, g)_{(i-1)q+j, (i'-1)q+j'} = \mathbb{1}_{\{g_{i,i'} = (h_i)_{j,j'} = 1\}}$ for all $i, i' \in [p]$ and $j, j' \in [q]$.

The next theorem specifies the representation of $G$-equivariant linear maps for an inductively-constructed $G$, by describing a basis of those linear maps. For a permutation group $G$ on $[m]$, let $\mathcal{E}(G) = \{M \in \mathbb{R}^{m \times m} : Mg = gM, \forall g \in G\}$, the vector space of $G$-equivariant linear maps, where each $g \in G$ is regarded as a permutation matrix. See Appendix B for the proof of Theorem 2.3.

**Theorem 2.3.** *Let $G, H$ be permutation groups on $[p]$ and $[q]$, and $\mathcal{B}(G), \mathcal{B}(H)$ be some bases of $\mathcal{E}(G)$ and $\mathcal{E}(H)$, respectively. Then, the following sets form bases for $G \oplus H$, $G \otimes H$, and $H \wr G$:*

$$\mathcal{B}(G \oplus H) = \{A \oplus \mathbf{0}_q, \mathbf{0}_p \oplus B : A \in \mathcal{B}(G), B \in \mathcal{B}(H)\}$$
$$\cup \{\mathbf{1}_{O \times (p+O')}, \mathbf{1}_{(p+O') \times O} : O \in \mathcal{O}(G), O' \in \mathcal{O}(H)\};$$
$$\mathcal{B}(G \otimes H) = \{A \otimes B : A \in \mathcal{B}(G), B \in \mathcal{B}(H)\};$$
$$\mathcal{B}(H \wr G) = \{A \otimes \mathbf{1}_{O' \times O''} : A \in \mathcal{B}(G), A_{i,i} = 0 \text{ for } i \in [p] \text{ and } O', O'' \in \mathcal{O}(H)\}$$
$$\cup \{I_O \otimes B : B \in \mathcal{B}(H), O \in \mathcal{O}(G)\}.$$

*Here $\mathbf{0}_m$ is an everywhere-zero matrix in $\mathbb{R}^{m \times m}$, and $\mathbf{1}_{R \times S}$, $I_R$ are the matrices defined by $(\mathbf{1}_{R \times S})_{i,j} = \mathbb{1}_{\{i \in R, j \in S\}}$, $(I_R)_{i,j} = \mathbb{1}_{\{i = j, i \in R\}}$ whose shapes are defined by the context in which they are used. Here, $\mathbf{1}_{O \times (p+O')}, \mathbf{1}_{(p+O') \times O} \in \mathbb{R}^{(p+q) \times (p+q)}$, $\mathbf{1}_{O' \times O''} \in \mathbb{R}^{q \times q}$, $I_O \in \mathbb{R}^{p \times p}$. Also, $\mathcal{O}(G) = \{\{g(i) : g \in G\} : i \in [p]\}$ (i.e., the set of G-orbits), and $p + O = \{p + i : i \in O\}$.*

# 3 Symmetry-aware SATNet

In this section, we present SymSATNet, which abbreviates symmetry-aware SATNet. This variant is designed to operate when symmetries of a learning task are known a priori (via an algorithm or a domain expert). The proofs of the theorem and the lemma in the section are in Appendix C.

SymSATNet solves the optimisation problem of SATNet, but under the following assumptions:

**Assumption 3.1.** The optimisation objective $\langle C, V^T V \rangle$ in (1) as a map on $V = \mathbb{R}^{k \times n}$ is invariant under a permutation group $G$, whose action is of type in Section 2.2 (i.e., each $g \in G$ acts as a permutation on the columns of $V$).

Continuing our convention, we denote $P_g$ by $g$. One immediate consequence of Assumption 3.1 is

$$\langle C, V^T V \rangle = \langle C, (g \cdot V)^T (g \cdot V) \rangle = \langle C, (Vg^{-1})^T (Vg^{-1}) \rangle \quad \text{for all } g \in G \text{ and } V \in \mathbb{R}^{k \times n}.$$

The next theorem re-phrases this property of the optimisation objective as equivariance of $C$:

**Theorem 3.2.** *Let $C$ be a symmetric $n \times n$ matrix. Then,*

$$\langle C, V^T V \rangle = \langle C, (Vg^{-1})^T (Vg^{-1}) \rangle \tag{3}$$

*for all $V \in \mathbb{R}^{k \times n}$ and $g \in G$ if $C$ as a linear map on $\mathbb{R}^n$ is $G$-equivariant, that is, $Cg = gC$ for all $g \in G$. Furthermore, if $k = n$, the converse also holds.*

This theorem lets us incorporate the symmetries into the objective of SATNet, and leaves the handling of the diagonal constraints of SATNet. The next lemma says that those constraints require no special treatment, though, since they are already preserved by the action of any $g \in G$.

**Lemma 3.3.** *Let $V \in \mathbb{R}^{k \times n}$ and $g \in G$. Every column of $V$ has the $L_2$-norm $1$ if and only if every column of $Vg^{-1}$ has the $L_2$-norm $1$.*

Recall $\mathcal{E}(G) = \{M \in \mathbb{R}^{n \times n} : Mg = gM, \forall g \in G\}$ is the vector space of $G$-equivariant matrices. Let $\mathcal{E}(G)_s$ be the subset of $\mathcal{E}(G)$ containing only symmetric matrices. When $G$ is a permutation group constructed by direct sum, direct product, and wreath product, we can generate a basis $\mathcal{B}(G)$ of $\mathcal{E}(G)$ automatically using Theorem 2.3. Then, we can convert $\mathcal{B}(G)$ to an orthogonal basis of $\mathcal{E}(G)_s$ by applying the Gram-Schmidt orthogonalisation to $\{B + B^T : B \in \mathcal{B}(G)\}$. Let $\mathcal{B}(G)_s = \{B_1, \ldots, B_d\}$ be such an orthogonal basis of $\mathcal{E}(G)_s$.

SymSATNet is SATNet where the matrix $C$ in the optimisation objective has the form:

$$C = \sum_{\alpha=1}^{d} \theta_\alpha B_\alpha \tag{4}$$

for some scalars $\theta_1, \ldots, \theta_d \in \mathbb{R}$. Note that by this condition on the form of $C$, SymSATNet has only $d$ parameters $\theta_1, \ldots, \theta_d$, instead of $n \times m$ for some $m$ in the original formulation of SATNet. When the learning target has enough symmetries, $d$ is usually far smaller than $n^2$ or even $n$, and this reduction brings speed-up and improved generalisation.

The forward computation of SymSATNet is precisely that of SATNet, the repeated coordinate-wise updates until convergence, and the backward computation is the one of SATNet extended (by the chain rule) with a step backpropagating the derivatives $\partial l / \partial C$ to each $\partial l / \partial \theta_\alpha$ for $\alpha \in [d]$.[3]

We summarise SymSATNet below using the usual notation of SATNet ($\mathcal{I}, \mathcal{O}, V_\mathcal{I}, V_\mathcal{O}$, and $V$):

- The input is $V_\mathcal{I}$, the matrix of the input columns of $V$.
- The parameters are $(\theta_1, \ldots, \theta_d) \in \mathbb{R}^d$. They define the matrix $C$ by (4).
- The forward computation solves the following optimisation problem using coordinate descent, and returns $V_\mathcal{O}$, the matrix of the output columns of $V$:

$$\underset{V_\mathcal{O} \in \mathbb{R}^{k \times |\mathcal{O}|}}{\operatorname{argmin}} \langle C, V^T V \rangle \quad \text{subject to } \|v_o\| = 1 \text{ for } o \in \mathcal{O}.$$

- The backward computation computes $\partial l / \partial V_\mathcal{I}$ and $\partial l / \partial \theta_\alpha$ by (2) and the chain rule:

$$\partial l / \partial \theta_\alpha = (\partial l / \partial \operatorname{vec}(C)) \operatorname{vec}(B_\alpha) = -U^T W \operatorname{vec}(B_\alpha). \tag{5}$$

---

[3]SymSATNet is implemented based on the SATNet code [26] available under the MIT License.

# 4 Discovery of Symmetries

One obstacle for using SymSATNet is that the user has to specify symmetries. We now discuss how to alleviate this issue by presenting an algorithm for discovering candidate symmetries automatically.

The goal of our algorithm, denoted by SYMFIND, is to find a permutation group $G$ that captures the symmetries of an unknown learning target and is expressible by the following grammar: $G ::= \mathcal{I}_m \mid \mathbb{Z}_m \mid \mathcal{S}_m \mid G \oplus G \mid G \otimes G \mid G \wr G$ for $m \in \mathbb{N}$. The $\mathcal{I}_m$ denotes the trivial group containing only the identity permutation on $[m]$, and $\mathbb{Z}_m$ denotes the group of cyclic permutations on $[m]$, each of which maps $i \in [m]$ to $(i + n) \bmod m$ for some $n$. The $\mathcal{S}_m$ is the group of all the permutations on $[m]$. The last three cases are direct sum, direct product, and wreath product (see Definition 2.2). They describe three ways of decomposing a group $G$ into smaller parts. Having such a decomposition of $G$ brings the benefit to recursively and efficiently compute a basis of $G$-equivariant linear maps.

The design of SYMFIND is based on our empirical observation that a softened version of symmetries often emerges in the parameter matrix $C$ of the original SATNet during training. Even in the early part of training, many entries of $C$ share similar values, and there is a large-enough group $G$ with $Cg \approx gC$ for all $g \in G$, which intuitively means that $G$ captures symmetries of $C$. Furthermore, we observed, such $G$ often consists of symmetries of the learning target. This observation suggests an algorithm that takes $C$ as input and finds such $G$ expressible in our grammar or its slight extension.

The input of SYMFIND is a matrix $M \in \mathbb{R}^{m \times m}$. As previously explained, when SYMFIND is called at the top level, it receives as input the parameter $C$ of SATNet learnt by a fixed number of training steps. However, subsequent recursive calls to SYMFIND may have input $M$ different from $C$. Then, SYMFIND returns a group $G$ in our grammar and a permutation $\sigma$ on $[m]$, together defining a permutation group on $[m]$:

$$\text{SYMFIND}(M) = (G, \sigma);$$
$$\text{grp}(G, \sigma) = \{\sigma \circ g \circ \sigma^{-1} : g \in G\},$$

where $\circ$ is the composition of permutations. When $G$ is decomposed into, say $G_1 \oplus G_2$, the $\sigma$ specifies which indices in $[m]$ get permuted by $G_1$ and $G_2$. Once top-level SYMFIND returns $(G, \sigma)$, we construct $\mathcal{B}(\text{grp}(G, \sigma))_s$, as in Section 3 with a minor adjustment with $\sigma$.[4]

**Algorithm 1** SYMFIND with a threshold $\lambda > 0$

1: **Input:** $M \in \mathbb{R}^{m \times m}$     **Output:** $(G, \sigma)$
2: **if** $\| \text{prj}(\text{grp}(\mathcal{S}_m, \text{id}_m), M) - M \|_F \leq \lambda$ **then**
3:     **return** $(\mathcal{S}_m, \text{id}_m)$
4: **end if**
5: $\mathcal{A} \leftarrow \{(\mathcal{I}_m, \text{id}_m)\}$
6: **if** $\| \text{prj}(\text{grp}(\mathbb{Z}_m, \text{id}_m), M) - M \|_F \leq \lambda$ **then**
7:     $\mathcal{A} \leftarrow \mathcal{A} \cup \{(\mathbb{Z}_m, \text{id}_m)\}$
8: **end if**
9: $(G', \sigma') \leftarrow \text{SUMFIND}(M)$;
10: $\mathcal{A} \leftarrow \mathcal{A} \cup \{(G', \sigma')\}$
11: **for** every divisor $p$ of $m$ **do**
12:     $(G'', \sigma'') \leftarrow \text{PRODFIND}(M, p)$;
13:     $\mathcal{A} \leftarrow \mathcal{A} \cup \{(G'', \sigma'')\}$
14: **end for**
15: $(G, \sigma) \leftarrow \text{argmin}_{(G,\sigma) \in \mathcal{A}} \dim(\mathcal{E}(\text{grp}(G, \sigma)))$
16: **return** $(G, \sigma)$

Algorithm 1 describes SYMFIND, where $\text{id}_m$ is the identity permutation on $[m]$, $\|\cdot\|_F$ is the Frobenius norm, $\dim(\mathcal{V})$ is the dimension of a vector space $\mathcal{V}$, and the Reynolds operator prj projects a matrix $M \in \mathbb{R}^{m \times m}$ orthogonally to the subspace of $G$-equivariant $m \times m$ matrices:

$$\text{prj}(G, M) = \frac{1}{|G|} \sum_{g \in G} gMg^T,$$

so that $\| \text{prj}(G, M) - M \|_F$ computes the $L^2$ distance between the matrix $M$ and the space $\mathcal{E}(G)$.

In the lines 2-4, the algorithm first checks whether $\mathcal{S}_m$ models symmetries of the input $M$ accurately. If so, the algorithm returns $(\mathcal{S}_m, \text{id}_m)$. Otherwise, it assumes that an appropriate group for $M$'s symmetries is one of the remaining cases in the grammar, and constructs a list $\mathcal{A}$ of candidates intially containing the trivial group $(\mathcal{I}_m, \text{id}_m)$. In the lines 6-8, the algorithm adds a pair $(\mathbb{Z}_m, \text{id}_m)$ to $\mathcal{A}$ if it approximates $M$'s symmetries well. In the lines 9-10, the algorithm calls the subroutine SUMFIND which finds $\text{grp}(G', \sigma')$ with $G' = \bigoplus_i G'_i$ that approximates $M$'s symmetries well. In the lines 11-14, the algorithm calls the other subroutine PRODFIND for every divisor $p$ of $m$. For each $p$, PRODFIND finds $\text{grp}(G'', \sigma'')$ with $G'' = G''_1 \otimes G''_2$ (or $G''_2 \wr G''_1$) that approximates $M$'s symmetries well, where $G''_1$ and $G''_2$ are permutation groups on $[p]$ and $[m/p]$. Finally, in the line 15, our SYMFIND picks a pair $(G, \sigma)$ from the candidates $\mathcal{A}$ with the strongest level of symmetries in the sense that the basis of $\text{grp}(G, \sigma)$-equivariant matrices has the fewest elements.

---

[4]We construct $\mathcal{B}(\text{grp}(G, \sigma)) = \{\sigma B \sigma^T : B \in \mathcal{B}(G)\}$, which is an orthogonal basis for $\mathcal{E}(\text{grp}(G, \sigma))$.

The subroutine SUMFIND clusters entries of $M$ as blocks since block-shaped clusters commonly arise in matrices equivariant with respect to a direct sum of groups. The other subroutine PRODFIND uses a technique [24] to exploit a typical pattern of Kronecker product of matrices, and detects the presence of the pattern in $M$ by applying SVD to a reshaped version of $M$. Each subroutine may call SYMFIND recursively. See Appendix D for the details.

## 5 Experimental Results

We experimentally evaluated SymSATNet and the SYMFIND algorithm on the tasks of learning rules of two problems, Sudoku and the completion problem of Rubik's cube. The original SATNet was used as a baseline, and both ground-truth and automatically-discovered symmetries were used for SymSATNet. For SYMFIND, we also tested its ability to recover known symmetries given randomly generated equivariant matrices. We observed significant improvement of SymSATNet over SATNet in various learning tasks, and also the promising results and limitation of SYMFIND.

**Sudoku problem** In Sudoku, we are asked to fill in the empty cells of a $9 \times 9$ board such that every row, every column, and each of nine $3 \times 3$ blocks have all numbers $1 - 9$. Let $A \in \{0, 1\}^{9 \times 9 \times 9}$ be the encoding of a full number assignment for the board where the $(i, j, k)$-th entry of $A$ is 1 if the $(i, j)$-th cell of the board contains $k$. In SATNet, we flatten $A$ to the assignment on the $n = 9^3$ boolean variables, and relax each variable into $\mathbb{R}^k$, resulting $V \in \mathbb{R}^{k \times n}$ in the objective of SATNet.

The rules of Sudoku have symmetries formalised by $G = (\mathcal{S}_3 \wr \mathcal{S}_3) \otimes (\mathcal{S}_3 \wr \mathcal{S}_3) \otimes \mathcal{S}_9$. Each of two $\mathcal{S}_3 \wr \mathcal{S}_3$ refers to solution-preserving permutations for rows and columns in Sudoku. The last $\mathcal{S}_9$ refers to permutations of the assigned numbers $1 - 9$ in each cell. See Appendix F for more information about the symmetry group for Sudoku.

To learn the rules of Sudoku using SymSATNet, we constructed a basis $\mathcal{B}(G)_s$ as explained in Section 3. It has 18 elements, which means that SymSATNet has 18 parameters to learn.

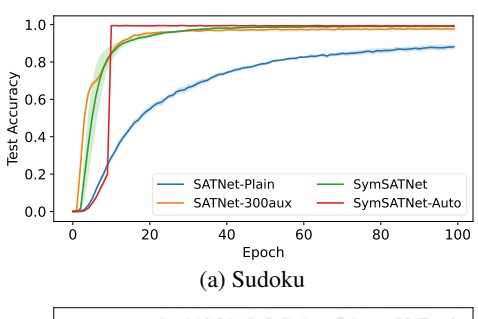

(a) Sudoku

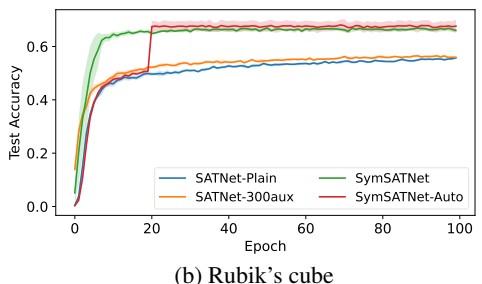

(b) Rubik's cube

Figure 1: Test accuracies over training epochs.

We used 9K training and 1K test examples generated by the Sudoku generator [21]. Each example is a pair $(V_{\mathcal{I}}, V_{\mathcal{O}})$ where the input $V_{\mathcal{I}}$ assigns 31-42 cells (out of 81 cells) and the output $V_{\mathcal{O}}$ specifies the remaining cells. SymSATNet was compared with SATNet-Plain without auxiliary variables, and SATNet-300aux with 300 auxiliary variables. We used binary cross entropy loss and Adam optimizer [16], with the learning rate $\eta = 2 \times 10^{-3}$ for SATNet-Plain and SATNet-300aux as the original work and $\eta = 4 \times 10^{-2}$ for SymSATNet. We measured test accuracy, the rate of the correctly-solved Sudoku instances by the forward computations. We reported the average over 10 runs with $95\%$ confidence interval.

Table 1: Best test accuracies during 100 epochs and average train times ($10^2$ sec). Additional times for automatic symmetry detection are also reported after $+$.

| MODEL | SUDOKU | | CUBE | |
|---|---|---|---|---|
| | ACC. | TIME | ACC. | TIME |
| SATNET-PLAIN | 88.1% | 48.0 | 55.7% | 1.8 |
| | ±1.8% | ±0.17 | ±0.7% | ±0.01 |
| SATNET-300AUX | 97.9% | 90.3 | 56.5% | 14.0 |
| | ±0.3% | ±0.68 | ±0.9% | ±0.12 |
| SYMSATNET | 99.2% | 25.6 | 66.9% | **1.1** |
| | ±0.2% | ±0.14 | ±1.2% | **±0.00** |
| SYMSATNET-AUTO | **99.5%** | **22.7** | **68.1%** | 3.4 |
| | **±0.2%** | **+0.14** | **±2.8%** | **+0.66** |
| | | **±0.35** | | ±0.19 |

The results are in Figure 1 and Table 1. (SymSATNet-Auto refers to a variant of SymSATNet that uses SYMFIND as a subroutine to find symmetries automatically, and will be described later in this section.) Our SymSATNet outperformed SATNet-Plain and SATNet-300aux. On average, its 100

epochs finished 2-4× faster in terms of wall clock than two alternatives. due to the reduced number of iterations and avoidance of matrix operations. See Appendix G for the efficiency of SymSATNet. Despite the speed up, the best test accuracy of SymSATNet (99.2%) was significantly better than SATNet-Plain (88.1%) and slightly better than SATNet-300aux (97.9%).

**Completion problem of Rubik's cube**    The Rubik's cube is composed of 6 faces, each of which has 9 facelets. We considered a constraint satisfaction problem where we are asked to complete the missing facelets of the Rubik's cube such that the resulting cube is solvable; by moving the cube, we can make all facelets in each face have the same colour, and no same colours appear in two faces. Let $A \in \{0,1\}^{6 \times 9 \times 6}$ be a colour assignment of Rubik's cube where the $(i,j,k)$-th entry has 1 if and only if the $j$-th facelet of the $i$-th face has colour $k$. We formulate the optimisation objective of SATNet for Rubik's cube using the relaxation of $A$ to $V \in \mathbb{R}^{k \times n}$ for $n = 6 \times 9 \times 6$.

This problem has symmetries formalised by $G = \mathcal{R}_{54} \otimes \mathcal{R}_6$ on $[n]$. Here $\mathcal{R}_{54}$ and $\mathcal{R}_6$ are permutation groups on $[54]$ and $[6]$, each of which captures the allowed moves of facelets, and the rotations of the whole cube. If a colour assignment $A$ is solvable, so is the transformation of $A$ by any permutations in $G$. See Appendix F for more information about the symmetries of this problem.

We generated a basis $\mathcal{B}(G)_s$ in three steps. We first created $\mathcal{B}(\mathcal{R}_{54})$ and $\mathcal{B}(\mathcal{R}_6)$ using the generators of each group [15]. Next, we combined them using Theorem 2.3 to get $\mathcal{B}(G)$, which was converted to a symmetric orthogonal basis $\mathcal{B}(G)_s$ via Gram-Schmidt. The final result has 48 basis elements.

We used a dataset of 9K training and 1K test examples generated by randomly applying moves to the solution of the cube. Each example is a pair $(V_{\mathcal{I}}, V_{\mathcal{O}})$ where $V_{\mathcal{I}}$ assigns colours to facelets except for two corner facelets, two edge facelets, and one center facelet, and $V_{\mathcal{O}}$ specifies the colours of those five missing facelets. In the test examples, only $V_{\mathcal{I}}$ is used. We trained SymSATNet, SATNet-Plain, and SATNet-300aux for 100 epochs, under the same configuration as in the Sudoku case.

The results appear in Figure 1 and Table 1. On average, the 100-epoch training of SymSATNet completed faster in the wall-clock time than those of SATNet-Plain and SATNet-300aux. Also, it achieved better test accuracies (66.9%) than these alternatives (55.7% and 56.5%). Note that unlike Sudoku, the test accuracy of SATNet-300aux was only marginally better than that of SATNet-Plain, which indicates that both suffered from the overfitting issue. Note also the sharp increase in the training time of SATNet-300aux. These two indicate that adding auxiliary variables is not so effective for the completion problem for Rubik's cube, while exploiting symmetries is still useful.

**Automatic discovery of symmetries**    To test the effectiveness of SYMFIND, we tested whether SYMFIND could find proper symmetries in Sudoku and Rubik's cube. We applied SYMFIND to the parameter $C$ of SATNet-Plain in $T$-th training epoch, where $T = 10$ for Sudoku and $T = 20$ for Rubik's cube. For Sudoku, SYMFIND always recovered the full symmetries with $G = (\mathcal{S}_3 \wr \mathcal{S}_3) \otimes (\mathcal{S}_3 \wr \mathcal{S}_3) \otimes \mathcal{S}_9$ in our 10 trials. For Rubik's cube, the group of full symmetries is $\mathrm{grp}(G, \sigma)$ for $G = ((\mathcal{S}_2 \wr \mathcal{S}_3) \oplus (\mathcal{S}_3 \wr \mathcal{S}_8) \oplus (\mathcal{S}_2 \wr \mathcal{S}_{12})) \otimes (\mathcal{S}_2 \wr \mathcal{S}_3)$. SYMFIND recovered all the parts except $\mathcal{S}_2 \wr \mathcal{S}_{12}$. Instead of this, the algorithm found $\mathcal{S}_{12} \otimes \mathcal{S}_2$ or $\mathcal{S}_3 \otimes \mathcal{S}_8$, or $\mathcal{S}_4 \otimes \mathcal{S}_6$ in our 10 trials. We manually observed that the entries of $C$ in the corresponding part were difficult to be clustered, violating the assumption of SYMFIND. This illustrates the fundamental limitation of SYMFIND.

To account for the limitation of SYMFIND, we refined the group $G$ of detected symmetries to a subgroup in an additional validation step, before training SymSATNet with those symmetries. In the validation step, we checked the usefulness of each part $G_i$ of the expression of $G$ in our grammar. Concretely, we rewrote $G$ only with the part $G_i$ in concern, where all the other parts of $G$ were masked by the trivial groups $\mathcal{I}_k$. After projecting $C$ with the masked groups using Reynolds operator, we measured the improvement of accuracy of SATNet over validation examples. Finally, we assembled only the parts $G_i$ that led to sufficient improvement. For example, if $G = (\mathbb{Z}_3 \otimes \mathcal{S}_4) \oplus \mathcal{S}_5$ is discovered by SYMFIND, we consider the parts $G_1 = \mathbb{Z}_3$, $G_2 = \mathcal{S}_4$, and $G_3 = \mathcal{S}_5$. Then, we construct three masked groups $G'_1 = (\mathbb{Z}_3 \otimes \mathcal{I}_4) \oplus \mathcal{I}_5$, $G'_2 = (\mathcal{I}_3 \otimes \mathcal{S}_4) \oplus \mathcal{I}_5$, and $G'_3 = (\mathcal{I}_3 \otimes \mathcal{I}_4) \oplus \mathcal{S}_5$, and measure the accuracy of SATNet with $C$ projected by each $G'_i$ over validation examples. If $G'_1$ and $G'_3$ show accuracy improvements greater than a threshold, we combine $G_1$ and $G_3$ to form $G' = (\mathbb{Z}_3 \otimes \mathcal{I}_4) \oplus \mathcal{S}_5$, which is then used to train SymSATNet.

We used 8K training, 1K validation, and 1K test examples to train SymSATNet with symmetries found by SYMFIND and the validation step. We denote these runs by SymSATNet-Auto. We took a group $G$ discovered by SYMFIND in $T$-th training epoch (with the same $T$ above) and constructed

its subgroup $G'$ via the validation step. SymSATNet was then trained after being initialised by the projection of $C$ with $G'$. The other configurations are the same as before.

As shown in Figure 1 and Table 1, SymSATNet-Auto performed the best for Sudoku (99.5%) and Rubik's cube (68.1%) better than even SymSATNet. During the 10 trials with Sudoku, SymSATNet-Auto was always given the full symmetries in Sudoku. For Rubik's cube, when SymSATNet-Auto was given correct subgroups (e.g., $((\mathcal{S}_2 \wr \mathcal{S}_3) \oplus (\mathcal{S}_3 \wr \mathcal{S}_8) \oplus (\mathcal{I}_4 \otimes \mathcal{I}_6)) \otimes (\mathcal{S}_2 \wr \mathcal{S}_3)$, $((\mathcal{S}_2 \wr \mathcal{S}_3) \oplus (\mathcal{S}_3 \wr (\mathcal{S}_2 \wr \mathcal{S}_4)) \oplus (\mathcal{I}_8 \otimes \mathcal{I}_3)) \otimes (\mathcal{S}_2 \wr \mathcal{S}_3)$), then it performed even better than SymSATNet. In two of the 10 trials, slightly incorrect symmetries were exploited, but it outperformed SATNet-Plain and SATNet-300aux. These results show the partial symmetries of subgroups derived by the validation step are still useful, even when they are slightly inaccurate.

**Robustness to noise** We tested robustness of SymSATNet and SymSATNet-Auto to noise by noise-corrupted datasets. We generated noisy Sudoku and Rubik's cube datasets where each training example is corrupted with noise; it alters the value of a random cell or the colour of a random facelet to a random value other than the original. We used noisy datasets with 0-3 corrupted instances to measure the test accuracy, and tried 10 runs for each dataset to report the average and 95% confidence interval. All the other setups are the same as before. Figure 2 shows the results. In both problems, SymSATNet was the most robust, showing remarkably consistent accuracies. SymSATNet-Auto showed comparable robustness to SATNet-300aux in noisy Sudoku, but outperformed the two baselines in noisy Rubik's cube.

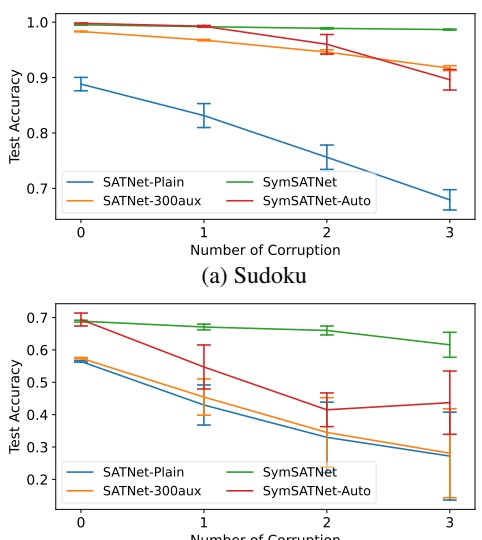

(a) Sudoku

(b) Rubik's cube

Figure 2: Best test accuracies for noisy Sudoku and Rubik's cube datasets.

Next, to show the robustness of SYMFIND, we applied it to restore permutation groups $G$ from noise-corrupted $G$-equivariant symmetric matrices $M$. We picked $(G_i, \sigma_i)$ for $i \in [4]$ where $\sigma_1, \sigma_2, \sigma_3$ are random permutations on [15], [30], [12], and $\sigma_4$ is the identity permutation on [8], and

$$G_1 = \mathbb{Z}_3 \oplus \mathbb{Z}_3 \oplus \mathbb{Z}_3 \oplus \mathbb{Z}_3 \oplus \mathbb{Z}_3, \quad G_2 = \mathcal{S}_3 \wr \mathcal{S}_{10}, \quad G_3 = (\mathcal{S}_3 \wr \mathcal{S}_3) \oplus \mathbb{Z}_3, \quad G_4 = \mathcal{S}_2 \otimes \mathcal{S}_2 \otimes \mathcal{S}_2.$$

Then, we generated $\mathrm{grp}(G_i, \sigma_i)$-equivariant symmetric matrices $M_i$ by projecting random matrices with standard normal entries into the space $\mathcal{E}(\mathrm{grp}(G_i, \sigma_i))_s$. Then, Gaussian noises from $\mathcal{N}(0, \omega^2)$ for $\omega = 5 \times 10^{-3}$ are added to $M_i$'s entries, and the resulting matrix $M_i'$ is given to SYMFIND.

For each $(G_i, \sigma_i)$, we repeatedly generated $M_i'$ and ran SYMFIND on $M_i'$ for 1K times, and measured the portion where SYMFIND recovered $(G_i, \sigma_i)$ exactly (full accuracy), and also the portion of cases where SYMFIND returned a subgroup of $(G_i, \sigma_i)$ which is not the trivial group $\mathcal{I}_m$ (partial accuracy). As Table 2 shows, the measured full accuracies were in the range of $60.3 - 93.5\%$, and the partial accuracies were in the range of $79.2 - 94.3\%$. These results show the ability of SYMFIND to recover meaningful and sometimes full symmetries.

Table 2: Full accuracies and partial accuracies of SYMFIND for given groups over 1K runs.

| GROUP | FULL ACC. | PARTIAL ACC. |
|---|---|---|
| $\bigoplus_{i=1}^{5} \mathbb{Z}_3$ | 76.6% | 79.2% |
| $\mathcal{S}_3 \wr \mathcal{S}_{10}$ | 60.3% | 79.9% |
| $(\mathcal{S}_3 \wr \mathcal{S}_3) \oplus \mathbb{Z}_3$ | 77.5% | 87.0% |
| $\mathcal{S}_2 \otimes \mathcal{S}_2 \otimes \mathcal{S}_2$ | 93.5% | 94.3% |

**Transfer learning** To test the transferability of SymSATNet, we generated Sudoku and Rubik's cube datasets with varying difficulties, where each dataset consisted of 9K training and 1K test examples. For SymSATNet-Auto, we split the 9K training examples into 8K training and 1K validation examples. We used three levels of difficulties for Sudoku and Rubik's cube (easy, normal, hard), based on the number of missing cells for Sudoku or missing facelets for Rubik's cube. The input part of Sudoku examples was generated with 21 (easy), or 31 (normal), or 41 masked cells (hard), and the input part of Rubik's cube examples was generated with 3 (easy), or 4 (normal), or 5 missing facelets (hard). For both problems, we used the training examples of easy or normal

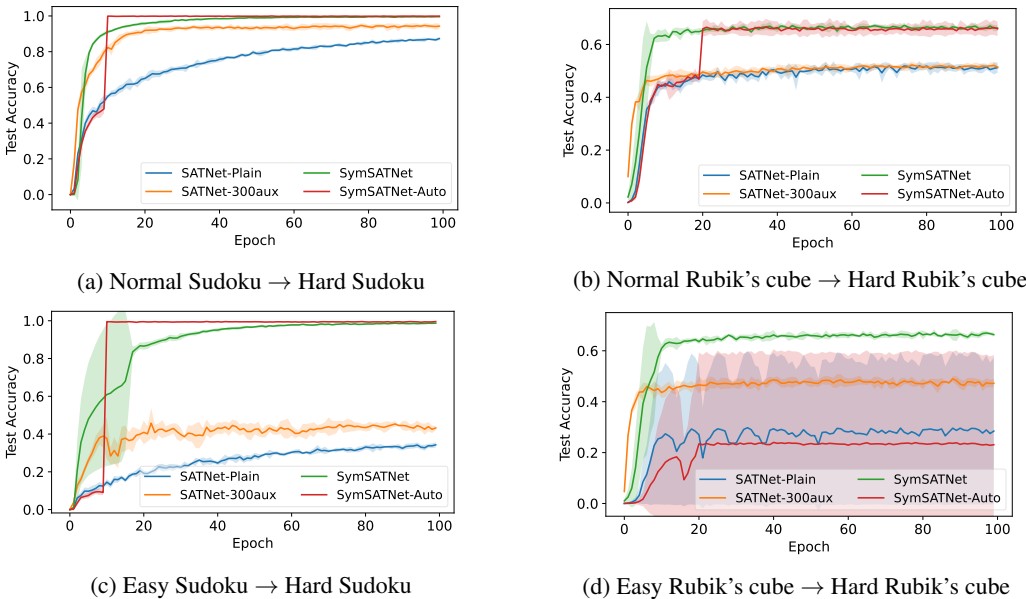

(a) Normal Sudoku → Hard Sudoku  (b) Normal Rubik's cube → Hard Rubik's cube

(c) Easy Sudoku → Hard Sudoku  (d) Easy Rubik's cube → Hard Rubik's cube

Figure 3: Transfer learning with various difficulties of training and test examples. For both problems, normal and easy examples were used to train, and hard examples were used to test each model.

datasets for training, and the test examples of hard datasets for testing. We repeated every task in this experiment five times. Here we report the average test accuracies and $95\%$ confidence interval.

Figure 3 shows the test accuracies throughout 100 epochs in the four types of transfer learning tasks. SymSATNet achieved the best result in the whole tasks; it succeeded in solving hard problems after learning from easier examples. As Figures 3a and 3c indicate, SymSATNet-Auto exploited the full group symmetries in Sudoku. For Rubik's cube, Figure 3b shows that SymSATNet-Auto achieved better performance over the baselines by finding partial symmetries. These results show the promise of SymSATNet and SymSATNet-Auto for learning transferable rules even from easier examples.

Note that for the easy Rubik's cube dataset, SymSATNet-Auto showed poor performance (Figure 3d). The poor performance comes from the violation of the assumption of SYMFIND; the group symmetries sometimes did not emerge in SATNet in this case. In three out of five trials with the easy Rubik's cube dataset, SATNet learnt nothing while producing the $0\%$ test accuracy, and SYMFIND returned the trivial group which equated SymSATNet-Auto with SATNet-Plain. In the remaining two trials, SATNet learnt correct rules, and SYMFIND and the validation step found correct partial symmetries, which led to the improved performance. These results exhibit the fundamental limitation of SymSATNet-Auto, whose performance strongly depends on the original SATNet.

## 6  Conclusion

We presented SymSATNet, that is capable of exploiting symmetries of the rules or constraints to be learnt by SATNet. We also described the SYMFIND algorithm for automatically discovering symmetries from the parameter $C$ of the original SATNet at a fixed training epoch, which is based on our empirical observation that symmetries emerge during training as duplicated or similar entries of $C$. Our experimental evaluations with two rule-learning problems related to Sudoku and Rubik's cube show the benefit of SymSATNet and the promise and limitation of SYMFIND. Although components of SYMFIND are motivated by the theoretical analysis of the space of equivariant matrices, such as Theorem 2.2, SYMFIND lacks a theoretical justification on its overall performance. One interesting future direction is to fill in this gap by identifying when symmetries emerge during the training of SATNet and proving probabilistic guarantees on when SYMFIND returns correct group symmetries.

**Acknowledgement**  This work was supported by the Engineering Research Center Program through the National Research Foundation of Korea (NRF) funded by the Korean Government MSIT (NRF-2018R1A5A1059921) and also by the Institute for Basic Science (IBS-R029-C1).

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
