# OpenReview forum: "Learning Symmetric Rules with SATNet"
_NeurIPS.cc/2022/Conference — NeurIPS 2022 Accept_

### Official Review · Reviewer_arDH · 2022-07-08

**Rating:** 8
**Confidence:** 4
**Soundness:** 4 excellent
**Presentation:** 3 good
**Contribution:** 3 good

**Summary:**

This paper proposes to incorporate permutation symmetries into the training objective of SATNet. The symmetry constraints reduce the number of parameters, and only requires slight modifications on the backward computation. The experiments show that adding symmetries could improve generalization and computation speeds. Furthermore, the paper also proposes an algorithm that finds symmetries automatically when the prior knowledge is not present.

**Questions:**

* Some neural combinatorial models, like NeuroSAT, enforces symmetries by designing specific network architectures. What do you think of the pros and cons of their approach and yours?
* Would you further explain the difference between SATNet and SATNet-300aux? Why does SATNet-300aux improve a lot more on sudoku not rubik's cube?
* What is the intuition that SymSATNet-auto performs better than SymSATNet when training and test are i.i.d but worse when they are not?
* Since I couldn't find a conclusion section, what do you think it's the main limitation of the paper and interesting future direction?

**Limitations:**

I didn't find a proper section in the paper that addresses the limitation.

Some I can think of:
* The approach can only handle permutation symmetries, not all symmetries.
* The approach only works with SATNet.


**Strengths And Weaknesses:**

Overall, this is a joyful reading for me. The paper is well-motivated. Exploiting symmetries is very important in combinatorial and logic reasoning problems. The proposed method is neat and fits nicely into the original framework of SATNet. The experiment results are also very promising. I'd recommend acceptance of this paper.

Below are some suggestions for the authors to further improve the paper:
* The proposed method is only able to handle permutation symmetries. There are more complicated symmetries which a permutation matrix is unable to specify. For example, SAT/UNSAT is invariant under transformations like unit propagation, variable elimination. However, the paper uses "symmetry rules" throughout, including the title. It would be good to be more careful about the term and explicitly point this out in the paper.
*  The paper lacks a proper conclusion section, which is a necessary place to summarize the paper, discuss limitations of the study and suggests future directions.
* Add transfer learning experiments. For combinatorial problems, it's often the case that training and test distributions are not i.i.d. Incorporating symmetries should improve the performance on transfer learning. The last experiment is conducted when test distributions are slightly perturbed from training. It would be good to include the set-up where training and test come from completely different distributions.
* Writing can definitely be improved. Some sentences are very wordy and long, which decreases the readability of the paper. One example is this sentence from line 27-30: "By symmetries, we mean transformations of candidate solutions of those rules, and by
rules having symmetries, we mean that the solution set of those rules is closed under the transformations of those symmetries."

---

> ### Author Response · Authors · 2022-08-02
> **Response to Reviewer arDH**
>
> We appreciate the helpful feedback and encouragement from the reviewer. We respond to the questions and comments that were not answered in our earlier responses.
>
> - [Q1] It would be better to point out that the paper focuses on the permutation symmetries rather than other transformations possible on SAT/UNSAT problems.
>   - [A1] Thank you for pointing this out. We revised the second paragraph in the introduction section and explicitly said that the paper focuses on the permutation symmetries.
>
> - [Q2] What are the main limitations of the paper and interesting future directions?
>   - [A2] The SymFind algorithm is based on the empirical observation that symmetries in target rules emerge in SATNet’s parameter matrix $C$ as clusters of $C$’s entries. This empirical observation is not justified theoretically, which is one limitation of our work. Lifting this limitation and answering theoretically when this symmetry emergence occurs is one interesting future direction. We plan to look at the gradient with respect to $C$ (computed during training by the backward pass of the original SATNet), and check when it carries information about symmetries of target rules. Also, analysing the formal properties of SymFind (e.g., the numerical bounds on the probability that SymFind returns correct group symmetries with respect to λ) would be another possible future direction. We added these discussions to the conclusion section of the paper.
>
> - [Q3] Add transfer learning experiments where training and test examples come from completely different distributions.
>   - [A3] Thank you for the great suggestion. We considered two options. The first is to mask cells or facelets such that their indices in the training examples and the test examples are disjoint (e.g., in Sudoku, we may mask only even-indexed rows in the training set, and only odd-indexed rows in the test set). The second option is to use the different levels of difficulties for the training examples and the test examples. We conducted the second experiment, and reported the results in Appendix J in the revised paper. The results describe better generalisation performance of SymSATNet over the baselines, and the promise and limitation of SymSATNet-Auto in the transfer learning tasks.
>
> - [Q4] Why does SymSATNet-Auto perform better than SymSATNet when training and test examples are i.i.d but worse when they are not?
>   - [A4] It is partly related to the discovered symmetry group $G$. In our non-iid experiment, SymSATNet-Auto is trained with noise-corrupted examples, so that its call to SymFind returns only a weak symmetry group $G$ (in the sense that the dimension of the space of $G$-equivariant matrices is large) and the projection of SATNet’s parameter $C$ to the $G$-equivariant space does not generalise well. In the iid case, the returned $G$ is strong (i.e. the space of $G$-equivariant matrices has a small dimension), and the projection generalises better. We think that the superior performance of SymSATNet-Auto comes from the smoother loss landscape of SATNet than that of SymSATNet, which is due to a larger number of trainable parameters in SATNet than SymSATNet. By training SATNet first and then projecting the trained SATNet’s parameter matrix $C$ into an equivariant matrix, SymSATNet-Auto enjoys both the smoother loss landscape of SATNet and the better generalisation of SymSATNet. We remark that after projection, SymSATNet-Auto behaves exactly like SymSATNet, using the equivariant matrices only. This is different from our earlier approach explained in our answer to Q2 of the reviewer 2aBF, which continues to use the full parameter matrix $C$ of SATNet after projection, and, in fact, applies projection periodically.

---

> > ### Comment · Reviewer_arDH · 2022-08-04
> > **After rebuttal**
> >
> > Thanks for the response. Authors' explanations clear up some of my confusions. This paper carefully studies the importance of exploring symmetries in discrete problems, and SymSATNet-Auto shows some promising evidence in the situation where the symmetries are unknown to the human. I think this paper can be impactful to some communities, such as, ML for combinatorial optimization and ML for mathematical reasoning. I have updated my score to reflect this.

---

### Official Review · Reviewer_NN1r · 2022-07-09

**Rating:** 6
**Confidence:** 3
**Soundness:** 2 fair
**Presentation:** 3 good
**Contribution:** 3 good

**Summary:**

The authors build on a differentiable constraint solver (SATNet, Wang et. al 2018) to propose a variant, SymSatNet, which exploits symmetries in the SAT problems in an attempt to improve the speed and accuracy of learning. This paper makes two primary contributions:
1. SymSatNet, their symmetry-aware formulation of the original SATNet model, which allows the model to exploit known (user-specified) symmetries and therefore constrain the optimization problem;
2. SymFIND, a method for discovering candidate symmetries based on approximate symmetries in the original SATNet model during training. This algorithm attempts to alleviate the domain-knowledge necessary to specify symmetries in advance in order to exploit the SySatNet formulation.



**Questions:**

1. Is it possible to better characterize when the SymFind algorithm should discover a candidate symmetry (eg. based on the training data it has observed?); and how many symmetries it can find with respect to the full ground truth symmetries?
2. Is it possible for SymFind to propose incorrect symmetries? What will happen if this is the case empirically, or in the worst case?
3. How does the SymSATNet model (and, for that matter, SATNet) perform with respect to a symbolic Sat solver on these domains?
4. How does SymFind perform with respect to existing symmetry-breaking techniques for SAT solving?
5. How does SymSATNet, the baseline SATNet, and the automatic symmetry finding problem scale with the number of constraints/propositions to be solved?


**Limitations:**

 My main point here is already largely covered in the sections above, so I will be brief: the authors do discuss empirical limitations of their automatic symmetry finding algorithm. However, I found that their current evaluation, on the Sudoku domain, left me with more questions than answers: when the model discovers only some symmetries, whether it can discover incorrect symmetries, and why we should expect this model to work in general (as well as its scalability as a symmetry finding algorithm.) I do believe this can be addressed, which is why I am hoping to see some empirical or theoretical response from the authors.


**Strengths And Weaknesses:**

Overall summary of comments: I am interested in this paper, which I think presents both compelling empirical evidence for the base formulation of their model, and which presents an interesting algorithm for discovering symmetries in constraint problems as part of training a differentiable constraint solver. However, as I discuss below, in many ways this paper appears preliminary, and I think there is significant room for improvement in its presentation, motivation, and empirical validation. I think this is a promising paper, and it’s unfortunate that the NeurIPS review process doesn’t provide an opportunity for revision. I’ll try to provide broad and specific points for improvement below, and perhaps we can work with the authors as part of the response to effectively provide revised portions of the paper and additional experiments during the rebuttal.

**Strengths**:
- Empirical demonstration of SymSATNet algorithm is compelling with respect to the SATNet baseline. The basic premise of this formulation is that exploiting symmetries in the constraint solving problem should significantly constrain the search space being implicitly optimized for by the differentiable solver. The authors demonstrate their model on two classically symmetric SAT problems -- Sudoku and Rubik’s cube solving -- and show that their model both improves on test-time accuracy and generally appears to learn more quickly than the original SATNet model.
- Empirical demonstration of the SymFind (automatic symmetry finding) algorithm is also promising on the Sudoku and Rubik’s cube domains, where the model appears to generally discover useful symmetries comparable to those specified in the base (known symmetries) model.
SymFind algorithm is original and seems like a potentially novel and interesting method for discovering symmetries in SAT problems that could be of interest both to the original target audience of the SATNet paper (deep-learning practitioners looking to incorporate ‘logic-aware components’ into neural models) and to classical constraint solving problems (as a means for discovering symmetries.) However, I believe this algorithm is presented in quite a preliminary way, as I discuss below, and its presentation, verification, and validation could be improved to convincingly demonstrate what exactly the authors are onto here.
- Noisy SAT solving results are interesting as a bonus.

**Weaknesses**:
- SymFind algorithm should either be better theoretically motivated, empirically validated, or both. I think this is the place where author response could yield the greatest differential on this paper, so I am noting it first. The automatic symmetry discovery, in my opinion, is the most interesting part of this paper, and also seems important to the original goals of the SATNet model. I actually think that it is fine that the model sometimes discovers only partial symmetries -- this would seem to be expected in general. However, I was not clear on either the theoretical behavior of this algorithm (is it possible for the model to overpropose symmetries that seemed to exist during training, but actually were not true symmetries in general ? Can the expected behavior be characterized with respect to the training curriculum, eg. the ‘percentage’ of symmetries observed? Could we construct adversarial training curricula for this model?); nor did the evaluation seem sufficiently robust. If the motivation for this approach is purely empirical -- that is, it seems that softened symmetries emerge in the parameter matrix C during training -- then that should be demonstrated using a much larger scale and targeted experiment -- eg. rather than focusing on the Rubik’s cube or Sudoku problems, I would want to see something more like “sampling large numbers of SAT proposition problems with explicit symmetries/permutations, sampling possible assignments, and demonstrating what ‘percentage’ of symmetries this model discovers in general.” Does that make sense?
Performance with respect to existing symbolic SAT solvers; and existing symmetry discovery methods? - - - The main evaluation is with respect to SATNet. This is, I think, also a weakness of the original SATNet paper, but seems important. How quickly can off-the-shelf SAT solvers solve the Sudoku and Rubik’s cube domains? Can existing symmetry-discovery algorithms designed for classical SAT problems robustly find symmetries from a similar training regime?
- Clarity and motivation of presentation. The introduction, in particular, assumes significant base knowledge of the original SATNet paper, and the paper can be quite dense and hard to read without first reading over the original. I think one or more other real-world examples in the introduction could help motivate this paper. A single well-made visualization might also greatly improve the presentation of the ‘symmetries’ problem, and the way in which this model attempts to discover approximate symmetries in the parameter matrix.
- As a small nit, it seems like the results section presentation of SymFind is out-of-date with respect to the rest of the paper: it uses an older name (SymSatNet-Auto) and references a ‘later’ presentation that in fact was moved before.

---

> ### Author Response · Authors · 2022-08-02
> **Response to Reviewer NN1r (Part 1/2)**
>
> We thank the reviewer for the detailed helpful feedback. We re-emphasise that although SATNet and SymSATNet can solve known constraints, they primarily aim at learning unknown rules or constraints from examples. This learning-constraint/rule/formula aspect is mostly absent in the work on SAT solvers mentioned by the reviewer. We respond to the reviewer’s questions and comments that were not addressed in our earlier responses.
>
> - [Q1] What will happen if SymFind proposes incorrect symmetries in the worst case?
>   - [A1] If SymFind proposes incorrect symmetries, training SymSATNet with those symmetries can be harmful; the test accuracy of the trained SymSATNet can be lower than that of the trained SATNet. To address this issue at least partially, we added a validation step after SymFind in our implementation; the step refines the group symmetries proposed by SymFind. This step picks only the useful parts among the proposed group symmetries, and thus alleviates the issue of using incorrect symmetries and damaging the training of SymSATNet. In our experiment with the Rubik’s cube problem, the proposed symmetries by SymFind had incorrect parts, which were fixed by the validation step. For details, see the second paragraph of the “Automatic discovery of symmetries” part in Section 5.
>
> - [Q2] Can you characterise the theoretical behaviour of the symmetry-discovery algorithm with respect to the training examples?
>   - [A2] Recall that the overall symmetry-discovery scheme consists of two steps. The first step is to train the original SATNet for some epochs, and store its learnt parameter matrix $C$, with the hope that $C$ retains the group symmetries of target rules to learn. The second step is the invocation of our SymFind algorithm on the stored $C$. SymFind detects the group symmetries in $C$ using the Reynolds operator, clustering, and SVD. The theoretical justification of the first step must answer when desired symmetries emerge in SATNet’s parameter $C$ in the early phase of training. Coming up with such a justification is highly nontrivial, and is a topic for future research. We think that a good starting point is to analyse when gradients computed by the backward pass of SATNet carry information about desired symmetries. For the second step, we point out that the key components of SymFind, namely, Reynolds operator, clustering, and SVD, are designed based on their nice theoretical properties, which are related to the search of a group $G$ meeting two conditions: (i) $Cg \approx gC$ for any $g \in G$ and (ii) $G$ has the strongest level of permutation symmetries among the groups satisfying (i). The Reynolds operator for a group $G$ is the projection function to the space of $G$-equivariant matrices; it maps a matrix to the closest $G$-equivariant matrix. In SymFind, we use this operator to measure the distance between the input matrix $C$ and the space of $G$-equivariant matrices, and in so doing, we ensure that the permutation group $G$ found by SymFind satisfies (i). Also, the clustering and SVD components (used in SumFind and ProdFind) are derived from the shapes of the basis elements of $G$-equivariant spaces, which are defined inductively in Theorem 2.3. These two components exploit the theorem to find a candidate $G$ satisfying i) and ii). Appendix D contains further details about this exploitation.
>
> - [Q3] How many symmetries can SymFind find with respect to the ground-truth symmetries?
>   - [A3] This depends on the hyperparameter $\lambda$, which acts as a threshold for deciding whether a given matrix $C$ (from a partially trained SATNet) has the symmetries of a candidate permutation group $G$, and which controls the tolerance of SymFind to noise. If $\lambda$ is close to $0$, SymFind becomes very strict to noise and returns a group $G$ where the input matrix is exactly in or very close to the space of $G$-equivariant matrices. If $\lambda$ is large, on the other hand, SymFind tends to be very permissive on noise and essentially returns the full permutation group $S_n$. In our experiments, we found an appropriate $\lambda$ by standard hyperparameter search, which in particular prevented SymFind from finding completely wrong symmetries. We also remark that SymFind discovered the full symmetries in the case of Sudoku.

---

> > ### Author Response · Authors · 2022-08-02
> > **Response to Reviewer NN1r (Part 2/2)**
> >
> > - [Q4] Are there theoretical notions with Symfind such as the number of symmetries, or indicators telling which symmetry is closer than others?
> >   - [A4] We used a dual notion to the number of symmetries in $G$, namely, the dimension of the space of $G$-equivariant matrices. This dimension can be computed efficiently and recursively for any symmetry group returned by SymFind. See Appendix B for the details (Claim B.4). We remark that the number of symmetries or elements in $G$, called order of $G$, often misbehaves. For instance, the order of the symmetry group of Sudoku is roughly 6.67e21, and this decreases to roughly 1.8e16 even if we only miss symmetries of permuting numbers, so that the “percentage” is not good with this notion because of the exponential scaling ($n!$ in the worst case). Note that the dimension of the space of $G$-equivariant matrices for any $G$ is bounded above by $n^2$, where $n$ is the number of variables.
> >   - We used the Reynolds operator, denoted prj in the paper, to measure the closeness or distance of a given matrix to the space of $G$-equivariant matrices. See Algorithm 1. The Reynolds operator can be computed efficiently using the basis of the space of $G$-equivariant matrices; see Theorem 2.3. The Reynolds operator is also used in the validation step, where the subgroups of the discovered symmetry group $G$ by SymFind are utilised to identify and refine incorrect parts of $G$.
> >
> > - [Q5] Could we construct an adversarial training model for SymFind?
> >   - [A5] SymFind would find wrong symmetries if such wrong symmetries emerge in SATNet’s parameter matrix C in the early phase of its training. Constructing a dataset that induces such C is a nontrivial task, and forms a nice future research direction. We remark that the standard notion of adversarial attack does not apply to SymFind directly. SymFind uses the Reynolds operator (which is Lipschitz) and L2-distance to decide whether the input matrix retains candidate group symmetries. As a result, a small perturbation to the input matrix does not make SymFind change its decision abruptly.
> >
> > - [Q6] Some real-world examples and well-made visualisation could improve the presentation of the paper.
> >   - [A6] The supplementary material and the appendix of our submission include some visualisation and animation. We plan to move the one in the appendix (Figure 4) to the main text in the final version of the paper.
> >
> > - [Q7] In the results section, it seems like the presentation of SymFind is out-of-date with respect to the rest of the paper: it uses an older name (SymSatNet-Auto) and references a ‘later’ presentation that in fact was moved before.
> >   - [A7] SymFind and SymSATNet-Auto are different. SymFind is an algorithm for finding symmetries from a given matrix, whereas SymSATNet-Auto refers to a version of the training algorithm of SymSATNet; SymSATNet-Auto not just uses SymFind as a subroutine to find symmetries in SATNet’s parameter matrix C automatically, but also initialises and updates the parameters of SymSATNet. To clarify this difference between the two, we revised the introductory sentence of SymSATNet-Auto in the paper.

---

> > > ### Comment · Reviewer_NN1r · 2022-08-09
> > > **Reponse Rebuttal**
> > >
> > > Thank you to the authors for the extremely detailed feedback to all reviewers, extensive discussion, and revision. I found the presentation to be highly improved.
> > >
> > > I agree with all of the points raised by the other reviewers as well -- both positive and negative. A good amount of my initial review was motivated by the desire to see a better theoretical grounding for this approach, and why it works, which I agree is highly nontrivial. But the discussion of [Q1] and [Q2] above were both extremely helpful in motivating this approach and highlighting the validation step. I also do agree with the central thrust of this response: that it is useful to be able to, essentially, induce new learned constraints, and codify them into exploitable symbolic rules.
> > >
> > > I've updated my score to reflect these responses. Thank you to everyone for a fruitful discussion period.

---

### Official Review · Reviewer_2aBF · 2022-07-11

**Rating:** 7
**Confidence:** 4
**Soundness:** 3 good
**Presentation:** 3 good
**Contribution:** 3 good

**Summary:**

This paper introduces SymSATNet, a modified version of SATNet that can find and exploit symmetries of a learnable logical formula.
Exploiting symmetries reduces the number of parameters and the complexity of the learning problem, which leads to faster training and better generalization.
Symmetries can either be specified a priori to integrate domain knowledge, or inferred from data with a novel 'SymFind' algorithm.

On a technical level, given a group of symmetries $G$ for the learnable formula, SymSATNet initially computes a basis for the vector space of $G$-equivariant linear maps.  It then parametrizes the learnable formula (represented as a matrix $C$) as a linear combination of such basis vectors, thereby ensuring that the learned formula obeys the given symmetries.

For discovering symmetries from data, the authors first train the standard SatNet for a small number of epochs on the available dataset. The SymFind algorithm is based on the observation that a 'softened' version of the true symmetries often emerges in the learned formula. To convert these 'soft' symmetries into 'hard' symmetries, the SymFind algorithm recursively finds the strongest combination $G$ of symmetry 'atoms' that still closely models the 'soft' symmetries exhibited by learned formula $C$.
Finally, the formula $C$ is projected onto the space of $G$-equivariant matrices (with the discovered symmetry group $G$) as a warmstart for training SymSATNet.

Empirically, the authors compare SymSATNet (with ground truth symmetries or symmetries discovered by the SymFind algorithm) to the standard SATNet baseline on a Sudoku and Rubik's cube task. In both problem setups, given a partial assignment of a set of discrete variables, the task is to predict the correct assignment of the remaining variables under a set of problem-specific rules. The SATNet/SymSATNet model maintains a learnable set of rules based upon which the assignment is completed, and this rule set is learned from a dataset of completed assignments.
In Sudoku, SymSATNet and SATNet (with auxiliary variables) achieve a close to perfect test accuracy, in Rubik's cube SymSATNet outperforms SATNet. In both experiments, the version of SymSATNet with discovered symmetries achieved similar performance as the one with ground truth symmetries, as it is shown that the discovered symmetries either match or closely resemble the ground truth.
Finally, the robustness to noisy/mislabeled data (for SymSATNet) and noisy formulas (for SymFind) is examined. In the presence of label noise, SymSATNet (with ground truth symmetries) outperforms the baseline in both problem setups, demonstrating the usefulness of known symmetries to correct for mislabeled data. Discovered symmetries demonstrate improved performance over the baseline in the Rubik's cube experiment, although exhibiting high variance.
Finally, the SymFind algorithm is shown to have some robustness to noise applied to a dataset of formulas generated from hand-crafted symmetries. This demonstrates the ability to still recover partial and sometimes complete symmetries from corrupted formulas.

**Questions:**

- The original SATNet is demonstrated to benefit from adding auxiliary variables. Is it also possible to include auxiliary variables in SymSATNet? If so this should also be explored in the experiments, otherwise, this limitation should be discussed.
- In Sudoku the problem reduces to only learning 18 parameters. While this clearly helps with generalization, I imagine it could also make training in general more difficult, as overparametrization typically allows for smoother training. As an alternative to explicitly parametrizing $C$ in an equivariant basis, have the authors also tried to parametrize the full matrix $C$ and then perform a differentiable projection onto the vector space with the desired symmetry properties on the forward pass? This could potentially result in similar generalization, but smoother training due to the increased number of learnable parameters.
- In SymSATNet with discovered symmetries, why is the initial SATNet training run stopped at different epochs for the two experiments? I.e. after 10 epochs for Sudoku and after 20 epochs for Rubik's cube. Is this a sensitive hyperparameter?
- See also questions raised in Strengths and Weaknesses.


**Limitations:**

1) It would be good to include a discussion on the limitations due to potentially non-usable auxiliary variables (in contrast to the original SATNet).
2) As pointed out in the Clarity section, it is currently not clear what the runtime of the SymFind algorithm is and how it scales. If the runtime of SymFind makes up for a relevant part of the training time of SymSATNet-Auto, a discussion of the runtime limitations (with considerations of scaling with dimensionality) would be necessary in my opinion.

If discussions of these remaining unclear limitations are added to the paper, I am willingly to increase the rating of the paper.

**Strengths And Weaknesses:**

#### Originality
The authors improve an existing method with the idea of incorporating the use of symmetries. As pointed out by the authors, using equivariances and symmetries in general in deep learning is a known approach, but to the best of my knowledge, this is the first time it is used for learning logical formulas from data. The described algorithm for discovering symmetries also appears to be novel.

#### Quality
Overall the paper has a good quality, it appears that many considerations were taken into account in the design of the method and the SymFind algorithm. It performs well in the presented set of experiments, speeding up training and improving the generalization capabilities.

Current weaknesses in quality:
- Related work:
	- Currently, it is not clear to me if some parts of the SymFind algorithm are inspired by existing works on discovering symmetries. It would be good to add how it conceptually differs from the listed related work.
- Experiments:
	- I find the experiments a bit too toy-ish. The Sudoku experiments seem to be too easy to make a strong point (already one of the baselines achieves an almost perfect test accuracy). While there is a clear performance improvement shown in the Rubik's cube experiment, it is still small-scale even compared to the visual sudoku presented in the original SATNet paper. Perhaps adding a visual component with a learnable neural network backbone in the experiment could make it more convincing.
	- Generally, it would be great to have more restarts of the experiments than 3 or 4, as the variance in some of the experiments is relatively large.
	- It would be great if the authors included also the train loss curves in the appendix for all experiments.
	- The validation step described in the paragraph starting at line 347 seems to be a vital component of the method. It would be good to show its effect in an ablation study in the appendix.


#### Clarity
Overall, the paper is well-written, with a clear presentation.
On a small note, it might be useful to add line numbers to the algorithm box.
Also, in table 1 I am not sure how the time per epoch was calculated for the SymSATNet-Auto. Does this take the runtime of the SymFind algorithm account? In any case it would be good to report the runtime of the SymFind algorithm separately from the time per epoch, as this is a potential limitation of the method.

#### Significance
The presented method is significant in my opinion. SymSATNet allows to incorporate strong prior knowledge into SATNet and should often find applications when the often-used SATNet is currently employed. With the presented SymFind algorithm, the method is also applicable in case of no available prior knowledge of symmetries, which further extends the applicability.

---

> ### Author Response · Authors · 2022-08-02
> **Response to Reviewer 2aBF**
>
> We would like to thank the reviewer for many helpful suggestions and great feedback. Most of the suggestions were already reflected in the revised paper or discussed in our earlier responses. Here we respond to the remaining questions.
>
> - [Q1] Do the running times in Table 1 take the SymFind algorithm into account?
>   - [A1] In the submitted version, we reported the total training time including the runtime of SymFind divided by the number of epochs. In the revised version, we separated out the runtime for automatic symmetry detection algorithms in Table 1.
>
> - [Q2] As an alternative to parameterising with an equivariant basis, did you try to parameterise the full matrix $C$ and then perform a differentiable projection onto the vector space with the desired symmetry properties on the forward pass? This could potentially result in similar generalisation, but smoother training due to the increased number of learnable parameters.
>   - [A2] We have tried a similar approach in the early stage of our work. We trained the full matrix $C$ via SATNet, thus fully parameterising the underlying matrix $S$ such that $C = S^T S$, and during training, we projected $C$ onto the space of $G$-equivariant matrices in the style of Frank-Wolfe. We tried a few policies on the projection-rate hyperparameter (i.e., how often we apply projection), but they did not work well. We observed that after the matrix $C$ is trained fairly close to the space of $G$-equivariant matrices, it started to deviate from the space in the later phase of training possibly due to overfitting; leaving the projected $C$ to the fully parameterised SATNet seemed harmful. Because of this empirical observation, our way of using differentiable projection (i.e., projecting regularly during SATNet’s training) was ruled out. But we agree that there is a trade-off between safe symmetry searching and learning, and the projection-based approach needs to be reconsidered in the future.
>
> - [Q3] In SymSATNet-Auto, why are the stopping epochs in the early phase of SATNet training different in two problems?
>   - [A3] In the Rubik’s cube case, we inspected SATNet’s parameter $C$ during training, and found that it was underfit in the 10th epoch; entries of $C$ were not similar enough and did not form clear clusters, making the discovery of symmetries challenging. Thus, we picked the 20th epoch to address this underfitting issue. We remark that we also tried later epochs than 20, but this generally showed poorer results; the trial led to weaker symmetries than those found with the 20th epoch. We conjecture that this was due to overfitting.
>
> - [Q4] Is it possible to include auxiliary variables in SymSATNet? If so, this should also be explored in the experiments.
>   - [A4] Yes, it is possible. If a permutation group $G$ captures group symmetries of a problem to be solved without auxiliary variables, then $G \oplus \mathcal{I}_m$ represents the extension of same symmetries for the $m$ auxiliary variables. Note the use of $\mathcal{I}_m$ here that acts trivially (i.e., permutes nothing) on those auxiliary variables. As you suggested, we are planning to conduct additional experiments for this scheme, and report the results in the final version of this paper.

---

> > ### Comment · Reviewer_2aBF · 2022-08-08
> > **Answer to author response**
> >
> > I first want to thank the authors for their detailed answer to the concerns raised by all the reviewers and the updated submission which already contains many of the requested changes.
> >
> > Regarding my original concerns, all my questions and comments regarding weaknesses were addressed in the rebuttal. Many of the suggestions were already included in the revised paper, including clarifying the relation to related work, increasing the number of restarts in experiments, including train loss curves, and separating the runtime of SymFind.
> > The authors also discussed my concern of small-scale experiments and the possibility of including auxiliary variables in SymSatNet. While I still think it would be great to have a more large scale experiment, I now agree that for the scope of this paper the comparison on Sudoku and Rubiks cube is sufficient, as the focus is on discovering symmetry groups which are sufficiently complex in these tasks.
> > For the additional experiments including auxiliary variables mentioned by the authors in the rebuttal, I would also love to see a case in which auxiliary variables also exhibit non-trivial symmetries (i.e. not the identity as proposed by the authors), although I am not sure how practical this scenario is.
> >
> > Regarding the concerns raised by the other two reviewers, I agree with most of the raised concerns/suggestions, all of which were discussed in the detailed author answer, and many are already included in the revised paper. I particularly like the transfer learning experiment suggested by reviewer arDH, which led to very interesting and promising results in the revised paper. I also share this reviewers overall evaluation and excitement regarding the paper.
> > Reviewer NN1r raises important concerns in a more critical evaluation, with the most important one being a lack of motivation for the success of the SymFind algorithm. I agree with the authors that it is difficult to derive theoretical guarantees given the strong dependence on the soft symmetries discovered by the original SATNet. The reviewer mentions that in this case of pure empirical motivation, the experiments should be more large scale to show how exactly the soft symmetries emerge during training of SATNet. I agree that a more in-depth (but not necessarily more large-scale) experimental evaluation of the soft symmetries would be great, as the automatic symmetry discovery is in my opinion the most exciting part of the paper, but it is completely built on this emergence of soft symmetries.
> >
> > One final suggestion regarding this remaining weakness of the paper would be to train SATNet and run the SymFind algorithm every epoch (or every few epochs). Then one can plot the percentage of ground truth symmetries discovered by the SymFind algorithm over the training epoch. This would make a strong point in visualizing the emergence of the soft symmetries, including the under- and overfitting mentioned in the authors response to my initial review. Such a plot could be included for both the Rubiks cube and the Sudoku experiment, including multiple restarts. I know that there is little remaining time for revision of the draft, however, I would also be happy to see some preliminary tabular version of these results (with potentially fewer restarts) and a more refined plot included in the final version of the paper. I am also not certain about the best metric to plot, potentially this could be the full accuracy as well as the partial accuracy, similar to the metrics reported in table 2.

---

> > > ### Author Response · Authors · 2022-08-09
> > > **Response to Reviewer 2aBF**
> > >
> > > We appreciate your reply with further comments and suggestions. We respond to the last point in the reply.
> > > - [Q5] Can you visualise the emergence of the soft symmetries, including the underfitting and overfitting mentioned in the author's response? Running the SymFind algorithm every epoch (or every few epochs) in SATNet might be an option.
> > >   - [A5] We conducted an experiment to detect the appearance and disappearance of soft symmetries in $C$ throughout the training epochs. Here are our preliminary results, including an anonymous link to the image of graphs, which we carried out last night. We plan to report the results with further experiments and more careful analysis in the final version of our paper.
> > >   - We measured how close $C$ is to the space $\mathcal{E}(G)$ by computing $\Vert \mathrm{prj}(G, C) - C \Vert_F$ (denoted by “projection error” below) where $G$ is the ground-truth permutation group for a given learning problem (e.g., Sudoku). We repeated 30 times for each problem to evaluate the projection errors throughout 100 training epochs. Please refer to [this figure](https://imgur.com/a/nzEyoWT) (an anonymous link) for the results. For Sudoku, in multiple cases out of 30 runs, the projection error hit the lowest point around the 5-10th epochs, and after these epochs, the projection error started to increase until certain epochs, so that the training ended with a high projection error. We did not observe any clear difference in the training loss (or the test loss) between these cases with high projection errors and the other cases (which showed low projection errors). We think this result suggests possible overfitting from the perspective of symmetry discovery in the original SATNet. For Rubik’s cube, in all of our trials, the projection error always decreased throughout the epochs, and no sign of overfitting was detected. Answering why this is the case is an interesting topic for future research. Also, in the very beginning of the training for both problems, the projection errors were not sufficiently small. By choosing proper stopping epochs in SymSATNet-Auto, we can avoid high projection errors, as we did (i.e., we picked the 10th epoch for Sudoku, and the 20th epoch for Rubik’s cube). Finally, we point out that there may be factors other than the projection error that influence the performance of our symmetry-discovery algorithm. As mentioned, we plan to carry out further experiments to check the existence and importance of such factors.

---

### Author Response · Authors · 2022-08-02
**Response to Common Questions (Part 1/3)**

We would like to thank the reviewers for giving helpful comments. We also appreciate their encouragement and acknowledgement for the novelty of our work. We plan to incorporate most of the reviewers’ suggestions into the paper, including the ones on presentation, as we will explain shortly. In the rebuttal, we will first describe the already-revised parts (which can be accessed in Openreview now) and respond to common concerns in the reviews. Then, we will move on to concerns specific to each review.

Here is the list of changes in the current revised version of the paper, which is available in Openreview. The changes are marked blue in the revised version.
- Further comparison between existing methods and our work is added to the related work part, which we also handle in the following response to the common questions.
- The line numbers are added to Algorithm 1.
- We repeated experiments more (totally 10 times) to reduce the variance of some results.
- We separately reported the runtime of automatic symmetry detection algorithms in Table 1.
- The conclusion part is added to summarise our contributions and suggest limitations and possible future work.
- Ablation study for the validation step in SymSATNet-Auto is added in Appendix I.
- Additional experiments on transfer learning are added. See Q3 of the reviewer arDH or Appendix J in the revised version of the paper.
- An overall visualisation of our work is added in the beginning of the appendix (Figure 3).
- Learning curves for the losses are added in Appendix K.
- Some typos and unnecessarily long sentences are fixed to improve readability.
- We added further analysis of computational costs of SymSATNet in Appendix F, which we also handle in the following response to the common questions.

These are our responses to the common questions in the reviews.
- [Q1] How do the computational costs of SATNet and SymSATNet scale with respect to the dimension of the input?
  - [A1] Let $n$ be the number of input variables, $k$ be the dimension for the values of those variables, and $m$ be the integer such that the parameter matrix $C$ of the original SATNet is of the form $S^T S$ for $S \in \mathbb{R}^{m \times n}$. The computational complexities of the forward and backward computations of the original SATNet are $O(nmk \times t)$ where $t$ is the number of iterations until convergence; recall that both forward and backward computations are done by iterative updates. For SymSATNet, the complexities of the forward and backward computations are $O(n^2d + (n^2k \times t))$ where $d$ is the number of basis elements of the space of equivariant matrices and $t$ is again the number of iterations until convergence. Note that $m$ crucially depends on the problem. For Sudoku, the original SATNet used $m = 600$ in a low rank structure and we used $m = n = 729$. We empirically observed that too small $m$ (e.g., $m = 300$ for Sudoku) causes underfitting, which results in substantial decrease of accuracy. We thus think that a well-chosen $m$ is $O(n)$ so that $m$ is interchangeable with $n$ in the analysis.
  - One factor that improves the runtime of SymSATNet over SATNet is $t$, the number of iterations until convergence. When we inspected the value of $t$ for SATNet and SymSATNet in our experiments, $t$ is smaller for SymSATNet in both problems due to the fast convergence ($t = 21.39$ versus $t = 19.1$ for Sudoku, and $t = 9.25$ versus $t = 7.3$ for Rubik’s cube on average). Besides the asymptotic analysis, in each iteration, SATNet requires two operations that cost $O(nmk)$, while SymSATNet requires only one operation that costs $O(n^2 k)$, which brings the improvement of the constant factor 2. This makes a meaningful difference on the scaling constants of the time complexities. See Appendix F for further analysis about the efficiency of SymSATNet compared with that of SATNet.

---

> ### Author Response · Authors · 2022-08-02
> **Response to Common Questions (Part 2/3)**
>
> - [Q2] What is the computational complexity of SymFind in terms of the dimension of the input?
>   - [A2] Suppose that an $n \times n$ matrix is given to SymFind as an input. The complexity of SymFind shown in Algorithm 1 is $O(n^{3 + \epsilon})$ for any arbitrarily small $\epsilon > 0$. When SumFind is called in the line 9 of Algorithm 1, it first spends $O(n^3)$ time for clustering, and then makes recursive calls to SymFind with $n_i \times n_i$ submatrices for $\sum n_i = n$. Each of these calls takes $O(n_i^{3 + \epsilon})$ time by induction, and the total cost of all the calls is $\sum_i O(n_i^{3 + \epsilon}) = O(n^{3 + \epsilon})$. Also, when ProdFind is called in the lines 11-12 of Algorithm 1, for each divisor $p$ of $n$, ProdFind performs SVD in $O(n^3)$ steps, and makes recursive calls to SymFind. The recursive calls here together cost $O(p^{3 + \epsilon / 2} + (n/p)^{3 + \epsilon / 2}) = O(n^{3 + \epsilon / 2})$ by induction. Thus, the loop in lines 11-14 costs $O(n^{3 + \epsilon / 2} d(n)) = O(n^{3 + \epsilon})$, where $d(n)$ is the number of divisors of $n$, since $d(n) = O(n^{\epsilon / 2})$ for any arbitrarily small $\epsilon > 0$. It remains to show that applying the Reynolds operator (in the lines 2 and 6) and counting the number of basis elements (in the line 15) take $O(n^{3 + \epsilon})$ steps. The former costs O(n^2) since it is actually an average pooling operation when we consider permutation groups. The latter costs at most $O(n)$ (when it is given $\mathcal{B}(\bigoplus_{i=1}^n \mathbb{Z}_1)$).
>
> - [Q3] Are some parts of SATNet, SymSATNet, or the SymFind algorithm inspired by or different from existing work on discovering symmetries (e.g., symmetry-breaking, NeuroSAT)?
>   - [A3] Let us first remind the reviewers that classical SAT solvers and NeuroSAT receive rules or constraints of a problem as an input, whereas SATNet and SymSATNet are designed to learn those rules from given training examples. Efficiently finding solutions to given rules or constraints is the focus of SAT solvers and NeuroSAT, while learning those rules with fewer examples in less time is the focus of SATNet and SymSATNet.
>   - The symmetry-breaking techniques for SAT solvers resemble our work in the sense that they prune isomorphic branches under permutation symmetries in assignment trees, so as to reduce the search space and the runtime of a SAT solver. Besides the use of symmetries for a different purpose (namely for constraint solving instead of constraint learning), these techniques detect hard symmetries of given CNF formulas, while our SymFind is designed to detect soft symmetries in parameter matrices C, which hold only approximately. Setting lambda in SymFind to 0 would make SymFind detect only hard symmetries and share the goal of those symmetry-breaking techniques. We also point out that intuitively parameter matrices C encode only restricted relaxed logical rules that track the interaction between two variables, but not among three or more variables. Parts of SymFind, such as the use of SVD, exploit this restriction. The symmetry-breaking techniques, on the other hand, target at handling more general formulas that specify interaction among three or more variables.
>   - As the reviewer arDH pointed out, NeuroSAT enforces permutation invariance and negation invariance by using a graph-based architecture. It constructs a graph of a given CNF formula, and predicts the satisfiability of the formula via a message-passing algorithm over the graph, which is known to be invariant with respect to the permutation of nodes in the graph. This is a different way of exploiting symmetries from what we do in SymSATNet. It does not reduce the size of the representation of a given formula using symmetries; instead, it uses a symmetry-preserving algorithm for inference on the formula. On the other hand, SymSATNet does reduce the size of rules represented as a matrix C directly. If SymSATNet had been designed similarly to NeuroSAT, it would have used the matrix C of SATNet without any restrictions, but would have come with symmetry-preserving algorithms for forward and backward computations. Exploring whether this hypothetical scenario or its combination with our approach in SymSATNet is possible is an interesting research direction. Specifically, NeuroSAT is shown to extrapolate to different classes of problems from those in the training set (e.g., it can be trained with random SAT problems to solve other SAT problems such as graph colouring). A positive answer from this exploration may lead to SymSATNet that can extrapolate.

---

> > ### Author Response · Authors · 2022-08-02
> > **Response to Common Questions (Part 3/3)**
> >
> > - [Q4] What makes the difference between the performance of SATNet and SATNet-300aux for Sudoku and Rubik's cube?
> >   - [A4] SATNet-300aux uses additional 300 auxiliary variables besides the original variables in order to encode the problem. The auxiliary variables are analogous to the slack variables in the SAT solving. The main difference is that these auxiliary variables are also a part of differentiable learning, and SATNet learns the relation between the original variables and the auxiliary variables, unlike hard-coded slack variables. This black-box approach in Sudoku worked fairly well and showed a high performance. However, in Rubik’s cube, it did not. In that case, both SATNet and SATNet-300aux showed a sign of overfitting even with a small number of missing labels. This suggests that Rubik’s cube is intrinsically harder than Sudoku, and SATNet and SATNet-300aux are unable to learn the logical rules accurately without additional prior information, such as information about group symmetries.
> >
> > - [Q5] Two problems in the experiments (Sudoku and Rubik’s cube) are easy or of small scales. It would be better to use much larger scale and targeted experiments (e.g., adding visual components, and using SAT problems with a large number of variables).
> >   - [A5] In our experiments, we chose Sudoku and the Rubik’s cube problem since they exhibit reasonably complex group symmetries, which we thought can test, in particular, the performance of our automatic symmetry detection. Although these problems are not large-scale examples, we think that the tried tasks with them are not easy. For instance, recall the performance drop of SATNet for noise-corrupted datasets (Figure 2), which indicates that the learning tasks on those datasets are  challenging. The experimental results on these tasks show the robustness of our SymSATNet and SymSATNet-Auto to noise. Also, our last experiment tests the ability of SymFind to recover the symmetries of general permutation groups.

---

### Meta-Review · Area_Chair_iNvg · 2022-08-26

**Recommendation:** Accept
**Confidence:** Less certain

**Metareview:**

This paper describes a refinement of SATNet that incorporates or finds symmetries in constrain satisfaction problems.  Experiments are given on Rubik's cube and Sudoku both of which exhibit significant symmetries (rotations and reflections of the cube and permutations of some rows and columns in Sudoku).  We have three reviews which are uniformly positive.  One weakness expressed is that the experiments seem toyish.  I also have that concern.  I would have to have see experiments on larger SAT problems before being convinced that there is something useful here.  But has the reviews are positive I will recommend acceptance.

**Award:**

No

---

### Decision · Program_Chairs · 2022-09-14

Accept